**Data Availability Statement:** All relevant data are within the manuscript and its Supporting Information files.

# Few SARS-CoV-2 infections detected in Newfoundland and Labrador in the absence of Public Health Laboratory-based confirmation

**Danielle P. Ings[1], Keeley M. Hatfield[1], Kathleen E. Fifield[1], Debbie O. A. Harnum[2], Kayla A. Holder[1], Rodney S. Russell[1], Michael D. Grant[1] ***

**1** Division of BioMedical Sciences, Faculty of Medicine, Memorial University of Newfoundland St. John's, NL, Canada, **2** Eastern Health Regional Health Authority, St. John's, NL, Canada

\* mgrant@mun.ca

## Abstract

### Objective

To assess the incidence of COVID-19 infection in the absence of a confirmatory test in persons suspecting they contracted COVID-19 and elucidate reasons for their belief.

### Methods

We recruited persons with a confirmed COVID-19 diagnosis and persons who believed they may have contracted COVID-19 between December, 2019 and April, 2021 into a study of immunity against SARS-CoV-2. An intake questionnaire captured their perceived risk factors for exposure and symptoms experienced, including symptom duration and severity. ELISA testing against multiple SARS-CoV-2 antigens was done to detect antibodies against SARS-CoV-2. No participant had received COVID-19 vaccination prior to the time of testing.

### Results

The vast majority of study subjects without Public Health confirmation of infection had no detectable antibodies against SARS-CoV-2. Suspected infection with SARS-CoV-2 generally involved experiencing symptoms common to many other respiratory infections. Unusually severe or persistent symptoms often supported suspicion of infection with SARS-CoV-2 as did travel or contact with travelers from outside Newfoundland and Labrador. Rare cases in which antibodies against SARS-CoV-2 were detected despite negative results of Public Health testing for SARS-CoV-2 RNA involved persons in close contact with confirmed cases.

### Conclusions

Broad public awareness and declaration of pandemic status in March, 2020 contributed to the perceived risk of contracting COVID-19 in Newfoundland and Labrador from late 2019 to

**Funding:** This work was supported by a Covid-19 rapid research funding opportunity grant (Funding Reference Number: VR1 – 173202) from the Canadian Institutes for Health Research awarded through the COVID Immunity Task Force to MDG, RSR and KAH. The funders had no role in study design, data collection and analysis, decision to publish, or preparation of the manuscript.

**Competing interests:** The authors have declared that no competing interests exist.

**Abbreviations:** BSA, bovine serum albumin; COVID-19, coronavirus disease-2019; ELISA, enzyme-linked immunosorbent assay; H1N1, influenza virus subtype H1N1; HKU1, human coronavirus HKU1; HRP, horseradish peroxidase; Min, minute; N, nucleocapsid; NL, Newfoundland and Labrador; OC43, human coronavirus OC43; OD, optical density; PCR, polymerase chain reaction; RBD, receptor binding domain; RNA, ribonucleic acid; S, full length spike protein; SARS-CoV-2, severe acute respiratory syndrome coronavirus 2; TMB, tetramethylbenzidine; WHO, World Health Organization.

April 2021 and raised expectation of its severity. Serological testing is useful to diagnose past infection with SARS-CoV-2 to accurately estimate population exposure rates.

## Introduction

Zoonotic introduction of severe acute respiratory syndrome coronavirus 2 (SARS-CoV-2) into the local population of Wuhan, China was highly publicized in late 2019, but the far-reaching consequences were yet unimagined [1]. International travel rapidly brought the virus to neighbouring countries, and by early 2020, SARS-CoV-2 had spread throughout most of Eurasia and to the western hemisphere [2, 3] (https://www.who.int/news/item/27-04-2020-who-timeline—covid-19). Constant news coverage created and maintained virtually universal awareness of SARS-CoV-2 and the associated illness termed coronavirus disease-2019 (COVID-19). On March 11, 2020, the World Health Organization (WHO) declared COVID-19 a pandemic (www.who.int/emergencies/diseases/novel-coronavirus-2019). Throughout early 2020, public anxiety in Canada steadily rose as the first wave of infections grew higher and wider (https://health-infobase.canada.ca/covid-19/epidemiological-summary-covid-19-cases.htm). The province of Newfoundland and Labrador (NL) fared relatively well in the first wave due in part to its island location and the decision many took to cancel travel plans scheduled during its late Spring break. The first documented case of SARS-CoV-2 infection in NL was reported on March 14, 2020. Despite one large cluster in the city of St. John's, the first wave of COVID-19 in NL was quickly brought under control and cases remained exceptionally low from May, 2020 until February, 2021 (www.gov.nl.ca/releases/covid-19-news).

Evaluating the role of immunity against SARS-CoV-2 involves investigating immune responses against SARS-CoV-2 across outcomes of exposure ranging from asymptomatic infection through hospitalization [4–7]. Therefore, beginning in August 2020, we recruited volunteers who either had recovered from a confirmed case of COVID-19 or believed they had been exposed to COVID-19, into a study of immune responses against SARS-CoV-2. The majority of persons volunteering did not have positive COVID-19 polymerase chain reaction (PCR) tests, but believed for various reasons that they may have acquired COVID-19. Of 217 persons without a positive COVID-19 PCR test that enrolled into our study, 208 showed no serological evidence of exposure to SARS-CoV-2. This illustrates the effectiveness of public health policies in identifying the majority of COVID-19 cases during the first wave of infections in NL. Conversely, the nine discordant cases, including 3 with negative COVID-19 PCR tests, demonstrate the limited window period for PCR-based confirmation of infection and indicate that accurate epidemiological assessment of case numbers requires supplementary testing [8–10]. We took the enrollment and testing of this cohort of COVID-19-negative volunteers as an opportunity to investigate what types of symptoms and encounters, in the context of a steady stream of information delivered through social and mainstream media, led people to suspect they had been infected with SARS-CoV-2. Our findings affirm that constant public attention had an impact on people's perception of how pervasive the spread of COVID-19 was from late 2019 through mid 2020 and of its case fatality rate across different age and risk groups [11]. A large fraction of those who thought they had COVID-19 related their belief to either the severity or protracted duration of symptoms, while others were alerted to specific symptoms such as loss of taste and smell or localized tissue discolouration. Certain types of encounters involving air travel were also interpreted as placing persons at risk for COVID-19. Psychophysiological impacts associated with pervasive experience of a growing pandemic

warrant further investigation [12–14]. The only clear indicator we observed for previous unconfirmed infection with SARS-CoV-2 in NL over this period was close contact with a PCR-confirmed case of COVID-19. Our study also illustrates the need for reliable COVID-19 antibody testing to identify persons previously infected with SARS-CoV-2 in the absence of a confirmatory PCR test and in addressing the immune status of persons suspecting they contracted COVID-19.

## Materials and methods

### Study participants and sample collection

This study was approved by the Newfoundland and Labrador Health Research Ethics Authority and carried out in accordance with recommendations of the Canadian Tri-Council Policy Statement: Ethical Conduct for Research Involving Humans. Eligible participants were recruited through news and social media, poster placement and word of mouth. In accordance with the Declaration of Helsinki, written informed consent was obtained for whole blood collection at three month intervals and a questionnaire addressing previous testing history and reasons for suspecting infection with SARS-CoV-2 was administered at study intake. Blood was drawn by forearm venipuncture into acid-citrate-dextrose preserved vacutainers. Plasma was collected by centrifuging whole blood for 10 minutes (min) at 500$g$. The upper acellular layer was removed and stored at -80 ˚C in small aliquots until testing.

### Serological testing

Plasma was diluted 1:100 in PBS containing 0.05% tween, 0.1% bovine serum albumin (BSA, Sigma Aldrich, St. Louis, MO, USA) and tested against recombinant proteins coated overnight at 50 ng/well in 50 μL carbonate/bicarbonate buffer (pH 9.2) onto 96 well Immunlon-2 ELISA plates (VWR Scientific, Mississauga, ON, Canada). Recombinant protein antigens for initial screening included the receptor binding domain (RBD) of SARS-CoV-2 spike protein (Sino-Biological, Wayne, PA, USA), full length SARS-CoV-2 spike protein (AcroBiosystems, Newark, DE, USA), SARS-CoV-2 nucleocapsid protein (SinoBiological) and BSA as a negative control. Plasma with reactivity against SARS-CoV-2 nucleocapsid protein in the absence of reactivity against SARS-CoV-2 spike protein RBD or intact spike protein was tested against OC43 and HKU1 β-coronavirus nucleocapsid proteins (SinoBiological) to assess whether exposure to common coronaviruses had induced antibodies cross reactive with SARS-CoV-2 nucleocapsid. Antigen-coated plates were washed 4 times with PBS-tween and blocked with 200 μL/well PBS with 1% BSA for 1 hour. Plates were again washed 4 times and diluted plasma added to duplicate wells at 100 μL/well for 90 min. Plates were washed 6 times and 100 μL/well of a 1:50,000 dilution of horseradish peroxidase (HRP)-conjugated polyclonal goat anti-human immunoglobulin G (Jackson ImmunoResearch Labs, West Grove, PA, USA) was added for 60 min. Plates were washed a further 6 times and 100 μL/well tetramethylbenzidine (TMB) substrate (Sigma Aldrich) was added. Plates were incubated for 20 min in the dark at room temperature, reactions stopped with 100 μL/well 2 N $H_2SO_4$ and optical density (OD) read at 450 nm on a BioTek synergy HT ELISA reader. High positive, low positive and negative control samples were run on each plate together with the unknown test samples. After subtracting background OD readings against BSA, OD readings against SARS CoV-2 spike RBD or intact spike protein more than 2 standard deviations above the mean OD of 40 plasma samples collected in NL prior to December, 2019 were considered indicative of a specific serological immune response against SARS-CoV-2. In the absence of significant reactivity against SARS-CoV-2 spike protein, selective reactivity against SARS-CoV-2 nucleocapsid compared

**Table 1. Categorization of subjects recruited into COVID-19 serology study in NL.**

|  | Detectable antibodies against SARS-CoV-2 | No detectable antibodies against SARS-CoV-2 |
| --- | --- | --- |
| PCR-confirmed COVID-19 | 44 | 2 |
| No PCR-confirmed COVID-19 | 6 | 196 |
| *Negative COVID-19 PCR test | 3 | 12 |

to the level of reactivity against OC43 or HKU1 nucleocapsid was considered indicative of a serological immune response against SARS-CoV-2.

## Results

### Serological assessment of SARS-CoV-2 exposure

In September, 2020, we began recruiting study subjects in NL who had previous confirmed infection with SARS-CoV-2 or who believed they may have contracted or been exposed to SARS-CoV-2. Of 263 persons recruited by April, 2021, 217 had never had a positive COVID-19 PCR test (Table 1). To determine which, if any, of these persons had previously been infected with SARS-CoV-2, we tested plasma collected at study entry against recombinant SARS-CoV-2 antigens using an in house ELISA validated with known positive controls and 40 plasma samples collected in NL before October, 2019. Virtually all previously confirmed cases of COVID-19 (44/46) had detectable antibodies against SARS-CoV-2 spike, spike RBD and nucleocapsid as illustrated with representative samples in Fig 1A. Eight persons without a confirmed COVID-19 PCR test (three who were negative when tested and five never tested) had detectable antibodies against SARS-CoV-2 spike RBD or full length spike (Fig 1B). Twelve persons without a confirmed COVID-19 PCR test had no detectable antibodies against SARS-CoV-2 spike RBD or full length spike protein, but had detectable antibodies against SARS-CoV-2 nucleocapsid and were initially classified as indeterminate (Fig 2A). As the nucleocapsid proteins of β-coronaviruses are relatively conserved, we compared plasma antibody binding to SARS-CoV-2, OC43 and HKU1 nucleocapsid proteins to test whether this reactivity against SARS-CoV-2 nucleocapsid was due to cross-reactive antibodies generated

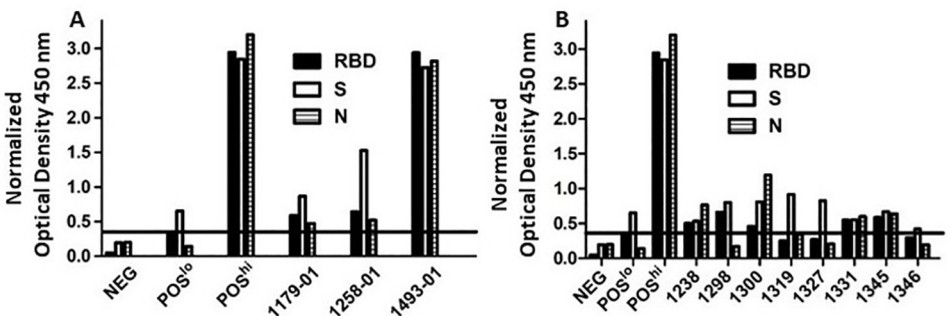

**Fig 1. Detection of SARS-CoV-2 specific antibodies.** Plasma samples collected from study volunteers were tested by ELISA against SARS-CoV-2 RBD, full length S protein and N protein. Panel A shows results with plasma from three representative subjects with confirmed COVID-19 infection along with negative, high positive and low positive control samples. Panel B shows results from eight study subjects without a PCR-based confirmatory test. The horizontal line represents the cutoff value for positivity based on reactivity with S or RBD for plasma samples collected from healthy control volunteers prior to October, 2019. The cutoff value was set at two standard deviations above the mean optical density of the plasma samples collected from healthy control volunteers prior to October, 2019. Optical density readings were normalized against the level of the high positive standard run on each ELISA.

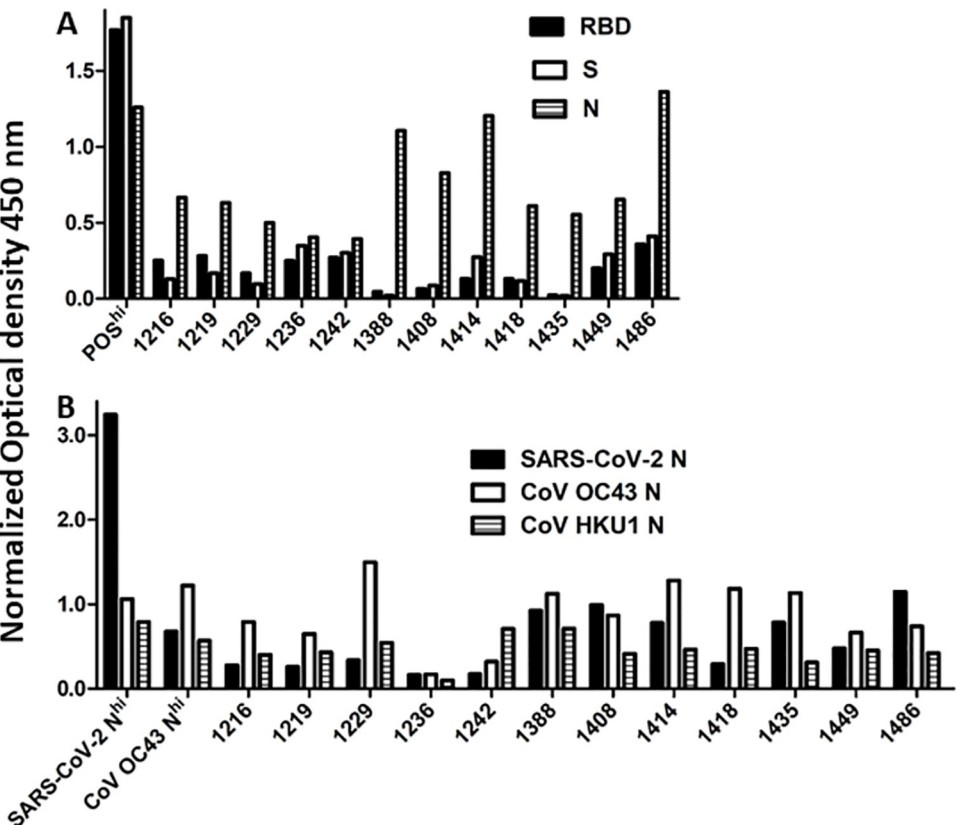

**Fig 2. Assessment of selective antibody binding to SARS-CoV-2 nucleocapsid (N) protein.** Twelve subjects with antibody levels against SARS-CoV-2 spike RBD and S that were below the cut off value for positivity had detectable antibodies against SARS-CoV-2 N protein (A). To test whether this was due to cross-reactivity against common β-coronavirus N proteins, we compared antibody binding to SARS-CoV-2, OC43 and HKU1 N proteins (B). Optical density readings were normalized against the level of the high positive standard run on each ELISA.

against common β-coronavirus nucleocapsids [15, 16]. Just 1/12 indeterminate cases showed clear selective binding to SARS-CoV-2 nucleocapsid indicative of previous infection with, or exposure to SARS-CoV-2 (Fig 2B). The only COVID-19 cases confirmed by Public Health testing where antibodies against SARS-CoV-2 were absent were one person with asymptomatic infection and no direct contact with any other confirmed case and one person with Crohn's disease who was receiving the immune modulators azathioprine and monoclonal anti-tumor necrosis factor-α.

## Reasons for suspecting SARS-CoV-2 infection

For study subjects with no confirmation of infection by Public Health and no detectable antibodies against SARS-CoV-2, self-reported symptoms or other reasons for suspecting SARS-CoV-2 infection were extracted from questionnaires, categorized and tabulated to assess and compare their prevalence (Table 2). Age and sex distribution is also reported therein. Many more subjects than had PCR-confirmed infection with SARS-CoV-2 or detectable antibodies against SARS-CoV-2 suspected they had contracted COVID-19 based on their experience of various symptoms across a range of severity and duration (Fig 3). Symptoms reported were primarily those common to numerous respiratory infections (cough, fever, fatigue and shortness of breath most prevalent) together with headache, myalgia, sore throat, rhinitis, nausea,

**Table 2. Age, sex and symptoms of persons suspecting exposure to SARS-CoV-2 without PCR-based diagnosis of active infection or serological evidence of past infection.**

|  | Male | Female |
|---|---|---|
| **Number of participants** | 76 | 131 |
| Age in years (mean ± standard deviation) | 49.6 ± 14.2 | 48.5 ± 13.8 |
| **Exposure** |  |  |
| Travel related | 53.2% | 41.2% |
| Close contact with confirmed case | 16.9% | 12.2% |
| **Symptoms experienced** |  |  |
| Cough | 57.1% | 66.4% |
| Fever | 58.4% | 62.6% |
| Fatigue | 46.8% | 58.0% |
| Shortness of breath | 35.1% | 32.8% |
| Headache | 29.9% | 32.1% |
| Myalgia | 24.6% | 30.5.% |
| Sore throat | 22.1% | 40.5% |
| Rhinitis | 22.1.% | 28.2% |
| Lost sense of taste or smell | 15.6% | 21.4% |
| Chest pain | 10.3% | 11.5% |
| Chest congestion | 10.3% | 6.1% |
| Nasal congestion | 7.8% | 11.5% |
| Diarrhea | 5.2% | 13.7% |
| Nausea/vomiting | 3.9% | 10.6% |
| Rash/discolouration | 2.6% | 3.1% |
| "Brain fog" | 2.6% | 1.5% |

vomiting and diarrhea (Table 2). This presumably reflects exposure to microbes circulating prior to introduction of SARS-CoV-2 into the human population. Shortness of breath, fatigue and loss of sense of taste/smell were the longest lasting symptoms reported. Fifty-seven persons stated that other members of their household were sick at the same time and 41 mentioned that they sought medical advice and or treatment from their physician. Anecdotal comments illustrating individual rationale for suspecting COVID-19 were also informative. For example, 99 persons described their symptoms as severe with 16 indicating they had never been so sick in their lives. Eight persons described their symptoms as worse than any flu they had ever experienced. Seventeen persons not meeting criteria for COVID-19 testing still believed they had contracted COVID-19 and 14 others stated they were sure they had COVID-19, regardless of negative PCR or antibody test results. Forty-three commented on the duration of illness, reporting they were experiencing symptoms more than two months after onset. Several people related their perceived risk of contracting COVID-19 to proximity to a coughing or otherwise symptomatic person on an airplane, proximity to other travelers in an airport or to work-related or social contact with international travelers. Of the nine persons with detectable anti-bodies against SARS-CoV-2 who had never had a positive COVID-19 PCR test, six reported air travel as the likely cause of exposure and four reported close contact with a known case of COVID-19.

## Discussion

Public awareness of COVID-19 was already high during reports of the initial outbreak in Wuhan and continued to grow with its rapid, widespread dissemination and progress to

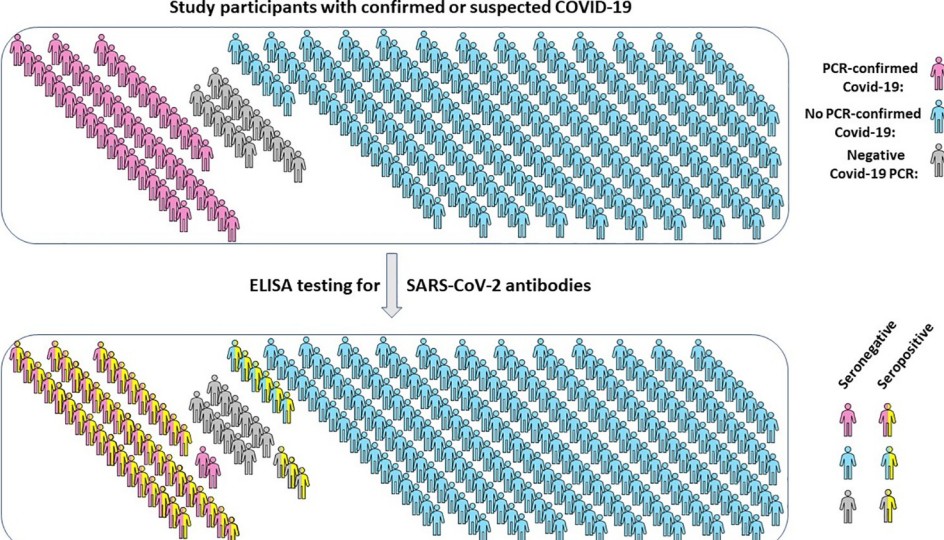

**Fig 3. Distribution of study subjects with confirmed or suspected COVID-19.** Of 263 individuals recruited, 46 had previous SARS-CoV-2 infection confirmed by PCR testing. All were subsequently tested by ELISA for antibodies against SARS-CoV-2 S and N proteins with results summarized as shown.

pandemic status [2, 3]. As of December, 2021, more than 275 million confirmed cases of COVID-19 and more than five million associated deaths have been reported worldwide (https://gisanddata.maps.arcgis.com/apps/dashboards/bda7594740fd40299423467b48e9ecf6). Depending upon host status, the severity of illness caused by SARS-CoV-2 infection ranges from mild, or even completely asymptomatic, to severe and lethal [17]. Clear risk factors for more severe infection include older age, certain co-morbidities and immune deficiencies [18, 19]. Certain lifestyle factors, including diet, can also contribute to risk for severe COVID-19 infection, both through a direct effect on physical well-being and an indirect influence on immune competence through modulating the gut microbiome [20, 21]. Several antiviral agents against SARS-CoV-2 have been developed, however, their efficacy is variable and often dependent upon intervening before the need for intervention is clear. A series of different biomolecules have been proposed as treatments and recently, 3 metabolites produced by components of the human microbiome were shown to be active against SARS-CoV-2 in vitro [22, 23].

Outbreaks of COVID-19 in long-term care facilities and among other vulnerable populations produce high case fatality rates that justify broad application of protective public health policies [24]. While appropriate concern about contracting COVID-19 helps support these policies, experiences related by participants in this study illustrate that a number of factors contribute to contextually exaggerated perceptions of both the risk of contracting COVID-19 and expectation of its severity. Continuous mainstream news and social media coverage with expert opinions raises awareness and knowledge, but what has been termed the "COVID infodemic" can also heighten perceptions of threat in some segments of the public [11, 25].

The initial outbreak of COVID-19 occurred in a setting densely populated with features thematic of pandemic fear. The prospect of a new lethal bird flu never strays far from public concern and the sudden appearance of a deadly new virus walking distance from a high level biohazard facility amid speculation of snakes, bats or pangolins from an exotic wet market infecting humans paints a psychologically unsettling picture [26]. Early news videos showed armies of cranes working to build entirely new hospitals to contain the sick. Conspiracy theories rapidly arose around suppression of information and the possibility of accidental or even

intentional creation of SARS-CoV-2 in a lab. As SARS-CoV-2 spread through Asia and Europe, mainstream news reported daily on explosive outbreaks and overwhelmed health care systems. When criteria were met for the WHO to label COVID-19 a pandemic, the public was well aware of vast numbers of hospitalizations and deaths worldwide and people were generally concerned for their own safety.

Against this disturbing backdrop, it's not surprising that many more persons than actually did contract COVID-19 suspected they might contract it and become very sick. The great pandemic toll of 1918 is deeply imprinted on society by regular reference and relatively recent pandemics such as human immunodeficiency virus, H1N1 influenza and the acute outbreaks from two previous emergent coronaviruses have updated fears. Popular cinematic dramatization of novel infections spreading as invisible invasions feed these fears, especially when reflected in an ongoing real life pandemic [27]. People in our study retrospectively indicated a belief they may have contracted COVID-19 in NL as far back as October 2019, in many cases citing direct or indirect contact with international travelers as the potential source. Although very rare, cases such as a single suspected infection in France prior to the outbreak in Wuhan encouraged people in our study with no known international exposure or other risk factors to consider the possibility they may have contracted COVID-19 in the fall of 2019 [28]. What appears to have been a severe respiratory illness circulating in NL in late 2019 through early 2020 led a number of our study participants to suspect they had contracted COVID-19. Common signs of a respiratory infection in the midst of a pandemic on everyone's mind made them naturally curious as to whether the symptoms they experienced were due to COVID-19.

The tendency for media and media consumers to focus on extremes in negative news affects fear in the population, which can be an appropriate adaptive response conferring protection, but can also increase psychological stress and anxiety [29]. Belief that the risk of contracting COVID-19 is high raises suspicion that symptoms common to many respiratory infections reflect COVID-19 and persons experiencing greater anxiety and stress report more COVID-like symptoms [30]. In addition, the belief that symptoms of COVID-19 are likely to be severe can influence perception of the severity of symptoms experienced, as illustrated by the nocebo effect [31]. Thus, persons believing they have contracted COVID-19 are more likely to experience severe symptoms during other infections.

Although serological testing is generally reliable in detecting past infection, an interesting group identified in this study were persons who spent considerable time in close contact with confirmed cases, but neither had a positive COVID-19 PCR test, nor developed detectable antibodies against SARS-CoV-2. Follow-up studies are underway to test for cross-reactive cellular immunity against common β-coronaviruses or other immune features that could confer partial resistance to COVID-19, as has been suggested in several recent studies [32, 33].

Another group that may be affected by elevated perceptions of the risk for and severity of SARS-CoV-2 infection are the COVID-19 "long haulers". At least some people suffering long-term effects attributed to COVID-19 have never had a confirmatory PCR test or been tested for antibodies against SARS CoV-2 [34]. Symptoms similar to those ascribed to the lasting effects of COVID-19 have been reported to occur following other viral infections, thus, it will be important to clearly distinguish those cases arising due to COVID-19 to properly characterize the syndrome and its etiology. It will also be important for evaluating duration of protection offered by vaccines or by recovery from COVID-19 to distinguish infection with SARS-CoV-2 from other infections with similar symptoms when a confirmatory PCR test is not obtained. As vaccines currently focus exclusively on SARS-CoV-2 spike protein, evaluating responses against other viral proteins sufficiently different from those of the previously more common coronaviruses will be key to determining if past, or now that vaccination is widespread, breakthrough infection has occurred.

## Conclusions

This study involved on a relatively small group of subjects recruited within a restricted geographic location over a short time period during an evolving pandemic. We can't report conclusively that COVID-19 infection doesn't occur in the absence of detectable seroconversion or state definitively that inordinate fear of COVID-19 was affected by media coverage between December 2019 and April 2021. While our results may not be generally representative of the Canadian or global population, they illustrate several key features of the public health response to COVID-19 in NL and of individual immune responses against SARS-CoV-2. Testing based on symptoms together with contact tracing was effective in identifying the vast majority of COVID-19 cases in the first wave of COVID-19 in NL and in limiting SARS-CoV-2 transmission. A relatively small number of COVID-19 infections went undiagnosed due to the timing of PCR testing. Virtually all cases of COVID-19 amongst our study participants resulted in readily detectable antibody responses persisting beyond 12 months against the neutralizing determinant of SARS-CoV-2 S protein. In agreement with a previous report, we detected antibodies induced by seasonal coronavirus infection that cross-reacted with SARS-CoV-2 N [16]. In rare cases with the absence of a specific antibody response against SARS-CoV-2 S protein, selectivity for SARS-CoV-2 N over the seasonal β-coronavirus nucleocapsids can signify previous exposure to SARS-CoV-2. Detection of antibodies against SARS-CoV-2 N alone without comparison to the level of antibodies against the common seasonal β-coronavirus nucleocapsids should not be considered diagnostic of past SARS-CoV-2 infection. Many more persons than actually contracted COVID-19, suspected they had, based on a variety of symptoms and experiences and especially perception of the severity or duration of symptoms. This indicates that perception of the likelihood of infection with SARS-CoV-2 can be inflated by intrinsic pandemic fear and immersion in widespread social and mainstream media coverage.

## Supporting information

**S1 Dataset.**
(XLSX)

## Acknowledgments

The authors would like to thank all study participants for their willingness to participate in this research study.

## Author Contributions

**Conceptualization:** Kayla A. Holder, Rodney S. Russell.

**Data curation:** Danielle P. Ings, Keeley M. Hatfield, Kathleen E. Fifield, Kayla A. Holder, Michael D. Grant.

**Formal analysis:** Danielle P. Ings, Michael D. Grant.

**Funding acquisition:** Kayla A. Holder, Rodney S. Russell, Michael D. Grant.

**Investigation:** Danielle P. Ings, Keeley M. Hatfield, Kathleen E. Fifield, Debbie O. A. Harnum, Kayla A. Holder, Rodney S. Russell, Michael D. Grant.

**Methodology:** Danielle P. Ings, Keeley M. Hatfield, Kathleen E. Fifield, Debbie O. A. Harnum, Kayla A. Holder, Rodney S. Russell, Michael D. Grant.

**Project administration:** Keeley M. Hatfield, Debbie O. A. Harnum, Kayla A. Holder, Rodney S. Russell, Michael D. Grant.

**Resources:** Michael D. Grant.

**Supervision:** Kayla A. Holder, Michael D. Grant.

**Visualization:** Danielle P. Ings.

**Writing – original draft:** Michael D. Grant.

**Writing – review & editing:** Rodney S. Russell, Michael D. Grant.

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
