## [Decision Letter · Decision Letter 0]

23 Nov 2021

PONE-D-21-35687Rare SARS-CoV-2 Infection in Newfoundland and Labrador in the Absence of Public Health Laboratory-based ConfirmationPLOS ONE

Dear Dr. Grant,

Thank you for submitting your manuscript to PLOS ONE. After careful consideration, we feel that it has merit but does not fully meet PLOS ONE’s publication criteria as it currently stands. Therefore, we invite you to submit a revised version of the manuscript that addresses the points raised during the review process.

We look forward to receiving your revised manuscript.

Kind regards,

Sanjay Kumar Singh Patel, Ph.D.

Academic Editor

PLOS ONE

Journal Requirements:

a) Did participants provide their written or verbal informed consent to participate in this study?

This work was supported by a Covid-19 rapid research funding opportunity grant (Funding Reference Number:  VR1 – 173202) from the Canadian Institutes for Health Research awarded through the COVID Immunity Task Force to MG, RR and KHo.   .  

Please state what role the funders took in the study.  If the funders had no role, please state: The funders had no role in study design, data collection and analysis, decision to publish, or preparation of the manuscript." 

5. Thank you for stating the following in the Funding Section of your manuscript: 

"This work was supported by a Covid-19 rapid research funding opportunity grant (Funding Reference Number:  VR1 – 173202) from the Canadian Institutes for Health Research awarded through the COVID Immunity Task Force to MG, RR and KHo."    

We note that you have provided funding information. However, funding information should not appear in the Funding section or other areas of your manuscript. We will only publish funding information present in the Funding Statement section of the online submission form. 

"This work was supported by a Covid-19 rapid research funding opportunity grant (Funding Reference Number: VR1 – 173202) from the Canadian Institutes for Health Research awarded through the COVID Immunity Task Force to MG, RR and KHo"

Reviewers' comments:

Reviewer's Responses to Questions

**Comments to the Author**

1. Is the manuscript technically sound, and do the data support the conclusions?

Reviewer #1: Yes

Reviewer #2: Yes

2. Has the statistical analysis been performed appropriately and rigorously? 

Reviewer #1: Yes

Reviewer #2: Yes

3. Have the authors made all data underlying the findings in their manuscript fully available?

Reviewer #1: Yes

Reviewer #2: Yes

4. Is the manuscript presented in an intelligible fashion and written in standard English?

Reviewer #1: Yes

Reviewer #2: Yes

5. Review Comments to the Author

Reviewer #1: The research article entitled, "Rare SARS-CoV-2 Infection in Newfoundland and Labrador in the Absence of Public Health Laboratory-based Confirmation" by Ings et al., assessed the incidence of COVID-19 infection in the absence of a confirmatory test in persons suspecting they contracted COVID-19, by ELISA testing against multiple SARS-CoV-2 antigens was done to detect antibodies against SARS-CoV-2. Authors have concluded that broad public awareness and declaration of pandemic status in March, 2020 contributed to the perceived risk of contracting COVID-19 in Newfoundland and Labrador from late 2019 to April 2021 and raised expectation of its severity. Altogether this is an important and timely research article, this reviewer has certain suggestions that would help produce a more comprehensive overview of the topic:

Comments:

1. Authors should provide limitations to their study.

2. At least one additional Figure (illustration) may be provided as to highlight the summary or prospect of this study.

3. To ensure the robustness and effectiveness of the risk model authors may support their finding with citing resent research articles to this manuscript.

4. The abbreviations should be cross validated in the manuscript (First define them fully followed by abbreviation) and one paragraph can be added for abbreviations.

Reviewer #2: The current research article entitled " Rare SARS-CoV-2 Infection in Newfoundland and Labrador in the Absence of Public Health Laboratory-based Confirmation" by Danielle Ings et al. has studied/surveyed using a questionnaire to capture their perceived risk factors for exposure and symptoms and multiple SARS-CoV-2 antigens were performed to detect antibodies against SARS-CoV-2. The inclusion criteria of the participants are confirmed COVID-19 diagnosis and the unvaccinated person believed to be in contact with COVID-19. The results suggested that most subjects had no detectable antibodies against SARS-CoV-2, and suspected individuals with COVID-19 had symptoms common to other respiratory infections. Only rare cases traveled from Newfoundland and Labrador have unusually severe or persistent symptoms despite detecting negative results of Public Health testing for SARS-CoV-2 RNA involved persons in close contact with confirmed cases. This study has some flaws which need some explanation from the author. However, the study addresses a research topic of great interest, but specific comments would help produce a more comprehensive overview of the topic:

Comments:

1. The English of the manuscript can be polished.

2. Line 219 in the manuscript looks misplaced.

3. First two paragraphs of the discussion section can be shortened and relevant information about mortality rate and various prevention approaches should be provided related to immunity and health i.e. doi: 10.1007/s12088-020-00908-0 and information regarding anti-COVID-19 Agents DOI: 10.1007/s12088-020-00893-4 can be provided.

4. The title of the manuscript should be modified. The manuscript does not provide substantial proof that unusually or persistent symptoms in persons in close contact with confirmed cases are due to rare SARS-CoV-2 variants.

5. In conclusion, in lines 320 to 324, the authors signify selectivity for SAR-CoV-2 nucleocapsid in detecting previous exposure to SARS-CoV-2. However, in the following line, the author also mentions that detecting antibodies against SARS-CoV-2 nucleocapsid alone cannot be considered diagnostic of past COVID-19 infections. Please explain then how the idea of the study can be justified when these two claims contradict each other.

6. In material methods section, the authors mention PBMCs are isolated and stored, but these PBMCs are not used in the manuscript. Therefore, please rewrite the material method section accordingly.

7. The author has represented his results in optical density at 450 nm. However, a ratio-based analysis of samples can be done for precise representation

8. Please mention how the cut-off value is determined for positivity based on reactivity with S or RBD in the manuscript.

---

## [Author Response · Author response to Decision Letter 0]

1 Jan 2022

Attached as response to reviewers.

---

## [Editor Report · Decision Letter 1]

10 Jan 2022

Few SARS-CoV-2 Infection in Newfoundland and Labrador in the Absence of Public Health Laboratory-based Confirmation

PONE-D-21-35687R1

Dear Dr. Grant,

We’re pleased to inform you that your manuscript has been judged scientifically suitable for publication and will be formally accepted for publication once it meets all outstanding technical requirements.

Kind regards,

Sanjay Kumar Singh Patel, Ph.D.

Academic Editor

PLOS ONE

---

## [Editor Report · Acceptance letter]

20 Jan 2022

PONE-D-21-35687R1 

Few SARS-CoV-2 infections detected in Newfoundland and Labrador in the absence of Public Health Laboratory-based confirmation 

Dear Dr. Grant:

I'm pleased to inform you that your manuscript has been deemed suitable for publication in PLOS ONE. Congratulations! Your manuscript is now with our production department. 

Kind regards, 

on behalf of

Dr. Sanjay Kumar Singh Patel 

Academic Editor

PLOS ONE